# The unified myofibrillar matrix for force generation in muscle

T. Bradley Willingham[1], Yuho Kim [1], Eric Lindberg[1], Christopher K. E. Bleck [1] & Brian Glancy [1,2]✉

Human movement occurs through contraction of the basic unit of the muscle cell, the sarcomere. Sarcomeres have long been considered to be arranged end-to-end in series along the length of the muscle into tube-like myofibrils with many individual, parallel myofibrils comprising the bulk of the muscle cell volume. Here, we demonstrate that striated muscle cells form a continuous myofibrillar matrix linked together by frequently branching sarcomeres. We find that all muscle cells contain highly connected myofibrillar networks though the frequency of sarcomere branching goes down from early to late postnatal development and is higher in slow-twitch than fast-twitch mature muscles. Moreover, we show that the myofibrillar matrix is united across the entire width of the muscle cell both at birth and in mature muscle. We propose that striated muscle force is generated by a singular, mesh-like myofibrillar network rather than many individual, parallel myofibrils.

[1] National Heart, Lung and Blood Institute, National Institutes of Health, Bethesda, MD 20892, USA. [2] National Institute of Arthritis and Musculoskeletal and Skin Diseases, National Institutes of Health, Bethesda, MD 20892, USA. ✉email: brian.glancy@nih.gov

The mechanisms underlying how muscle contraction generates the forces that propel human movement have been of fundamental interest for centuries[1–3], and much of our insight into muscle contractile function comes from our understanding of the highly patterned nature of muscle cell structure[4,5]. The long, cylindrical muscle cell comprises many repeating sarcomeres, which are understood to be arranged end to end in series to form tube-like myofibrils, which generate force longitudinally down the length of the cell[6,7]. The majority of muscle cell volume is thus considered to be made up of many individual, parallel myofibrils, and this textbook view has changed little since the 1800s[2,8,9]. Over 35 years ago, it was proposed that contractile force could also be transmitted laterally along the transverse axis of the muscle[10]. Since that time, the role of perpendicular cytoskeletal connections between individual myofibrils in lateral force transmission has been extensively studied[11–14]. However, despite the clear importance of desmin and other cytoskeletal proteins to normal muscle function, the precise mechanisms of lateral force transmission remain unknown[15–18].

Recent developments in 3D electron microscopy have led to fundamental changes in our understanding of how cellular architecture translates into cellular function[19–21]. Here we use high-resolution, 3D electron microscopy to demonstrate that striated muscle cells form a unified, nonlinear myofibrillar matrix connected across both the length and width of the cell by frequently branching sarcomeres. We observe myofibrillar networks in all muscle cells, although the frequency of sarcomere branching is regulated by muscle type and developmental stage. Additionally, we found that sarcomere branching can occur either through splitting of a single sarcomere into two or through a transfer of myofilaments between two adjacent sarcomeres, and that the frequency of each branching mechanism also varies by muscle type. The contiguous nature of the myofibrillar matrix across both the length and width of the cell provides a direct pathway for both lateral and longitudinal active force transmission throughout the muscle cell and argues against the presence of many individual myofibrils running the entire length of the muscle cell.

## Results

### Branching sarcomeres lead to a connected myofibrillar matrix.
To evaluate muscle contractile structure, we used focused ion beam-scanning electron microscopy (FIB-SEM)[19,20,22] to image mouse muscle volumes with 10 nm resolution in 3D (Supplementary Movie 1). Initially, we attempted to visualize individual myofibrils within the intact fast-twitch muscle cell by choosing a sarcomere from the muscle cross-section (Supplementary Fig. 1) and tracing its shape and that of each subsequent sarcomere connected in series through the z-disks along the length of the muscle. However, we found that sarcomeres often branched at an angle and resulted in a highly connected myofibrillar matrix rather than the textbook linear, tube-like myofibril structure (Fig. 1, Supplementary Movie 1). As a result, we modified our tracing routine to separate each myofibrillar segment within the myofibrillar matrix at the branching points (Fig. 1, Supplementary Movie 2). There was no apparent relationship between sarcomere branch points and specific sarcomeric regions such as the I-band (actin but no myosin present), A-band (actin and myosin overlap), or H-zone (myosin but no actin present) as sarcomere branch points could be seen in all three regions as well as just before and after the z-disks (Fig. 1). Additionally, sarcomere branches did not appear to alter the normal spatial relationships between the sarcomere and mitochondria (Supplementary Movie 3) or the sarcotubular systems (Supplementary Movie 4).

To exclude the possibility that sarcomere branches were caused by muscle damage or tissue fixation artifacts, we closely examined

muscle cell structures for abnormalities. Mitochondria, sarcoplasmic reticulum, and t-tubules were all normal in appearance with no evidence of swelling, suggesting that physiological osmotic and ionic conditions were maintained through sample fixation (Supplementary Movie 1). Contractile structures were continuous and tightly packed together, and the visible presence of I-bands confirmed that the muscles were not hypercontracted. Moreover, z-disks were properly aligned in parallel for sarcomeres both with and without branching (Supplementary Movie 5, upper), and instances of Vernier displacement of z-disk sheets[23,24] (Supplementary Movie 5, lower) appeared to occur at a far lower frequency and much larger scale than branching of individual sarcomeres, further indicating that the presence of the myofibrillar matrix is not due to muscle damage or pathology. These data demonstrate that the contractile apparatus of healthy, fast-twitch skeletal muscle cells comprises a highly connected, nonlinear myofibrillar matrix linked together by branching sarcomeres.

### Developmental, fiber-type regulation of sarcomere branching.
To determine whether the myofibrillar matrix was limited to fast-twitch muscle fibers or ubiquitous to all muscle types, we next evaluated contractile structures from slow-twitch and cardiac muscle cells. Both slow-twitch muscle fibers, as characterized by their more elongated and tortuous sarcomere cross-sectional shapes, high mitochondrial content, and thicker z-disks compared to fast-twitch fibers[25,26] (Supplementary Movie 1), and cardiac myocytes (Fig. 2a) also comprise frequently branching sarcomeres, indicating the presence of a myofibrillar matrix in all mature striated muscle types. To evaluate whether myofibrillar matrix structures were the predominant contractile structure within muscle cells, we assessed how frequently branching myofibrillar structures occurred relative to the number of non-branching, individual myofibrils. By beginning with each parallel sarcomere at one end of the dataset (see Supplementary Fig. 1) and tracking its structure along the length of the muscle within the field of view (Supplementary Movie 1, left), we determined whether any of the serial sarcomeres branched or if a single tube-like myofibril was apparent. Nearly all myofibrillar structures (96.1 ± 1.4%, 99.4 ± 0.6%, and 94.4 ± 2.5% of fast-twitch, slow-twitch, and cardiac myofibrils, mean ± SE, $n = 4$, 3, and 5 muscle volumes and 468, 299, and 344 myofibrils, respectively) contained at least one branching sarcomere within the field of view, indicating that the myofibrillar matrix was the primary contractile structure in each adult muscle type (Fig. 2b). To further assess the degree of connectivity of the myofibrillar matrix, we measured how frequently sarcomere branching occurred for each muscle type. For every ten sarcomeres in series, 1.6 ± 0.2 sarcomeres contained branches in fast-twitch muscle ($n = 4$ muscle volumes, 193 myofibrils, 3477 sarcomeres), 2.2 ± 0.2 sarcomere branches occurred in cardiac muscle ($n = 5$ muscle volumes, 167 myofibrils, 2114 sarcomeres), and 4.3 ± 0.4 sarcomere branches occurred in slow-twitch muscle ($n = 3$ muscle volumes, 110 myofibrils, 1396 sarcomeres), significantly higher than either cardiac or fast-twitch muscle (Fig. 2c). These data indicate that while the myofibrillar matrix is ubiquitous across muscle types, the degree of connectivity is regulated according to muscle fiber type.

To further demonstrate that a unified myofibrillar matrix is the normal structure of mature skeletal muscle cells, we evaluated myofibrillar structures in FIB-SEM datasets of the vastus lateralis of three adult humans (<40 years of age) published as part of the Baltimore Longitudinal Study of Aging[27]. In human muscle, sarcomere branching occurred in 100% of the myofibrils we evaluated ($n = 60$ myofibrils, three muscle volumes). Moreover,

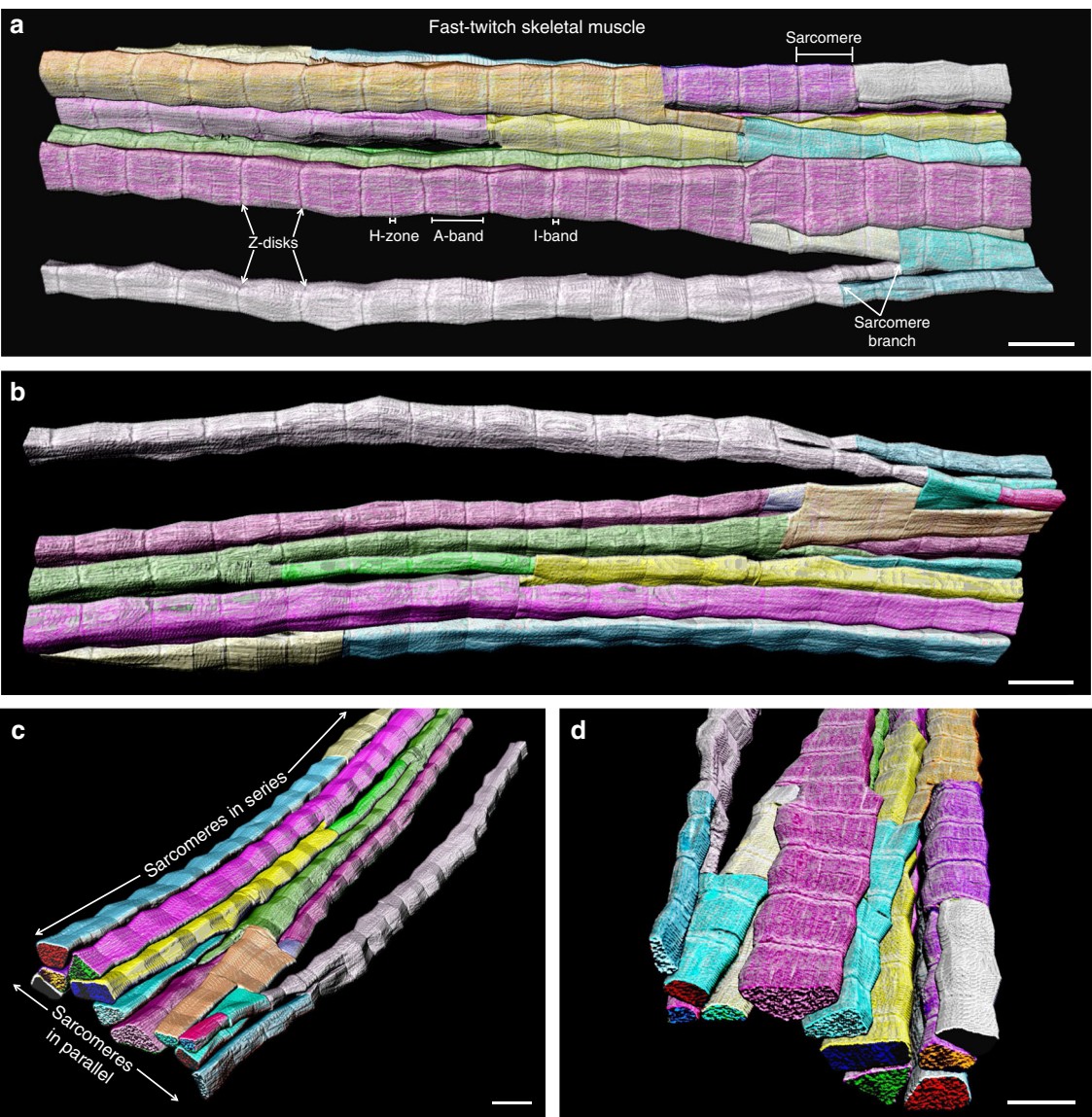

**Fig. 1 Muscle myofibrils form a highly connected matrix. a** 3D rendering of 115 directly connected sarcomeres within the myofibrillar matrix of a fast-twitch muscle. Individual colors represent different myofibrillar segments linked by branching sarcomeres. **b–d** 3D renderings at different perspectives highlight the nonlinearity and connections between myofibrillar segments. Representative of 7707 sarcomeres from four volumes from four mice. Scale bars, 2 μm.

analysis of human fast-twitch muscle sarcomere structures ($n = 906$ sarcomeres, three muscle volumes) revealed that sarcomere branching occurred at a frequency of $2.4 \pm 0.2$ branches per 10 sarcomeres (Supplementary Fig. 2) slightly higher than in the mature mouse fast-twitch muscle. These data demonstrate the presence of a highly branched myofibrillar matrix in stable, adult human skeletal muscle.

To investigate the formation of the myofibrillar matrix and the frequency of sarcomere branching during postnatal muscle development, we assessed contractile structures during the early (postnatal day 1 (P1)) and late (P14) (Fig. 2d) postnatal periods. Similar to adult muscles, $97.9 \pm 1.2\%$ of early postnatal myofibrillar structures ($n = 3$ muscle volumes, 150 myofibrils) had at least one branching sarcomere (Fig. 2b), demonstrating that the myofibrillar matrix was already the predominant contractile structure at the time of birth. However, the frequency of sarcomere branching in early postnatal muscles ($2.8 \pm 0.2$ branches per 10 sarcomeres, $n = 3$ muscle volumes, 150 myofibrils, 1401 sarcomeres, Fig. 2c) was higher than fast-

twitch muscles, but lower than slow-twitch muscles, suggesting that sarcomere branching is developmentally regulated. Unlike the adult and early postnatal muscles, only $70.0 \pm 3.1\%$ of late postnatal myofibrillar structures ($n = 3$ muscle volumes, 141 myofibrils) contained at least one branching sarcomere within our field of view (Fig. 2b), suggesting that sarcomere branching may be downregulated during the transition from the neonatal to adult contractile isoforms that occurs during this period[28]. Consistently, the frequency of sarcomere branching in the late postnatal muscles ($1.3 \pm 0.2$ branches per 10 sarcomeres, $n = 3$ muscle volumes, 141 myofibrils, 1796 sarcomeres, Fig. 2c) was the lowest of all muscle types assessed. Taken together, these data indicate multistage regulation of sarcomeric branching within the myofibrillar matrix during postnatal muscle development where the frequency of sarcomere branching is lowest during the period of greatest developmental muscle growth[29].

**Sarcomere branching occurs by splitting or content transfers.** To investigate the mechanisms of sarcomere branching, we

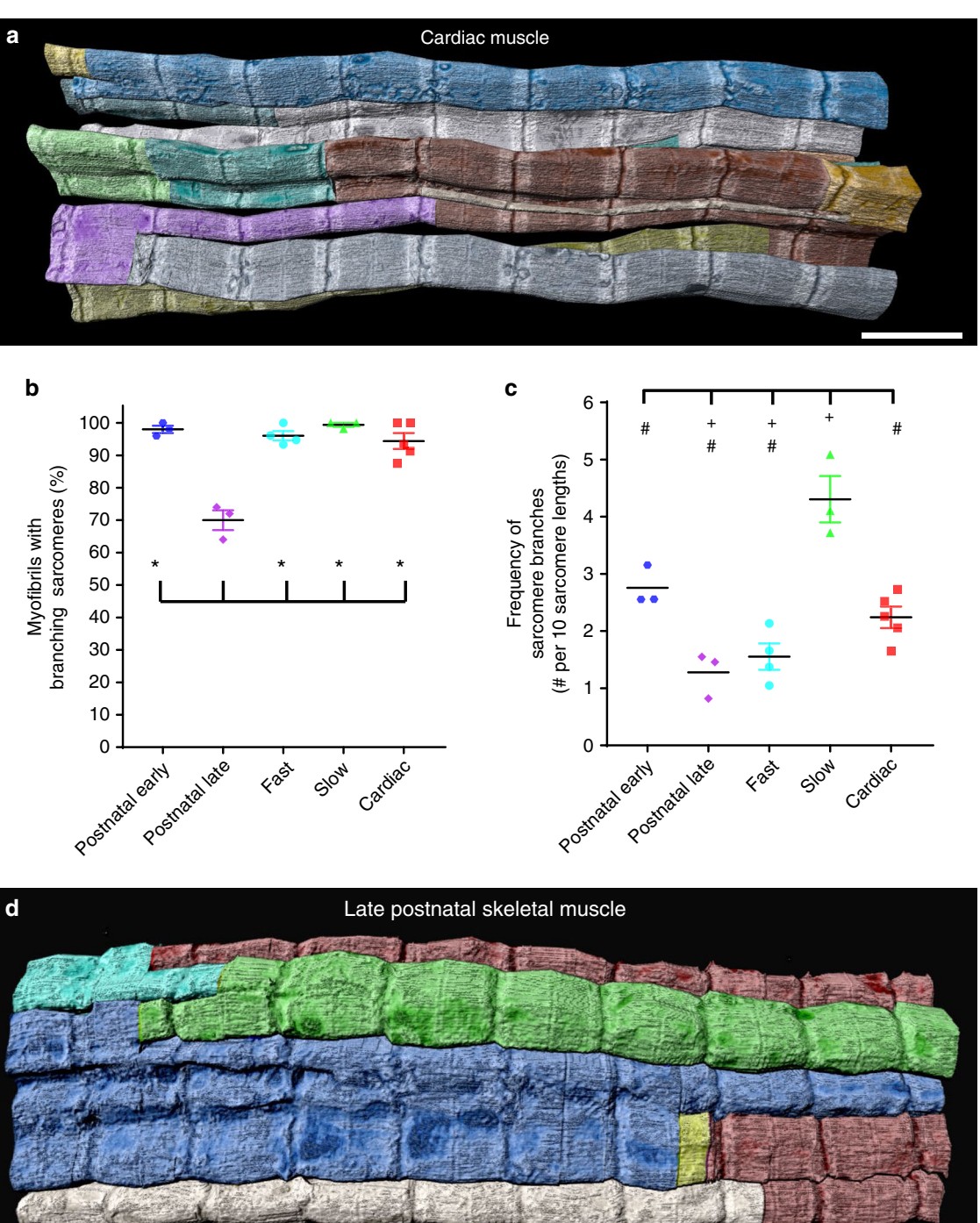

**Fig. 2 Sarcomere branching is extensive, but variable across muscle types and developmental stages. a** 3D rendering of 77 directly connected sarcomeres within the myofibrillar matrix of a cardiac muscle fiber. Individual colors represent different myofibrillar segments linked by branching sarcomeres. **b** Percentage of myofibrils with at least one branching sarcomere in different muscle fiber types. *N* values: postnatal early—141 myofibrils, 1796 sarcomeres, 3 muscles, 3 mice; postnatal late—150 myofibrils, 1401 sarcomeres, 3 muscles, 3 mice; fast twitch—468 myofibrils, 7707 sarcomeres, 4 muscles, 3 mice; slow twitch—299 myofibrils, 3990 sarcomeres, 3 muscles, 3 mice; cardiac—344 myofibrils, 4473 sarcomeres, 5 muscles, 5 mice. **c** Frequency of sarcomere branching across muscle type. *N* values: postnatal early—141 myofibrils, 1796 sarcomeres, 3 muscles, 3 mice; postnatal late—150 myofibrils, 1401 sarcomeres, 3 muscles, 3 mice; fast twitch—193 myofibrils, 3477 sarcomeres, 4 muscles, 3 mice; slow twitch—110 myofibrils, 1396 sarcomeres, 3 muscles, 3 mice; cardiac—167 myofibrils, 2114 sarcomeres, 5 muscles, 5 mice. **d** 3D rendering of 43 directly connected sarcomeres within the myofibrillar matrix of a late neonatal (P14) muscle fiber. Individual colors represent different myofibrillar segments linked by branching sarcomeres. Representative of three muscles from three mice. Scale bars, 2 μm. Plus sign (+): Significantly different (one-way ANOVA, *P* < 0.05) from postnatal early. Asterisk (*): Significantly different from postnatal late. Pound sign (#): Significantly different from slow. Values: mean ± SE.

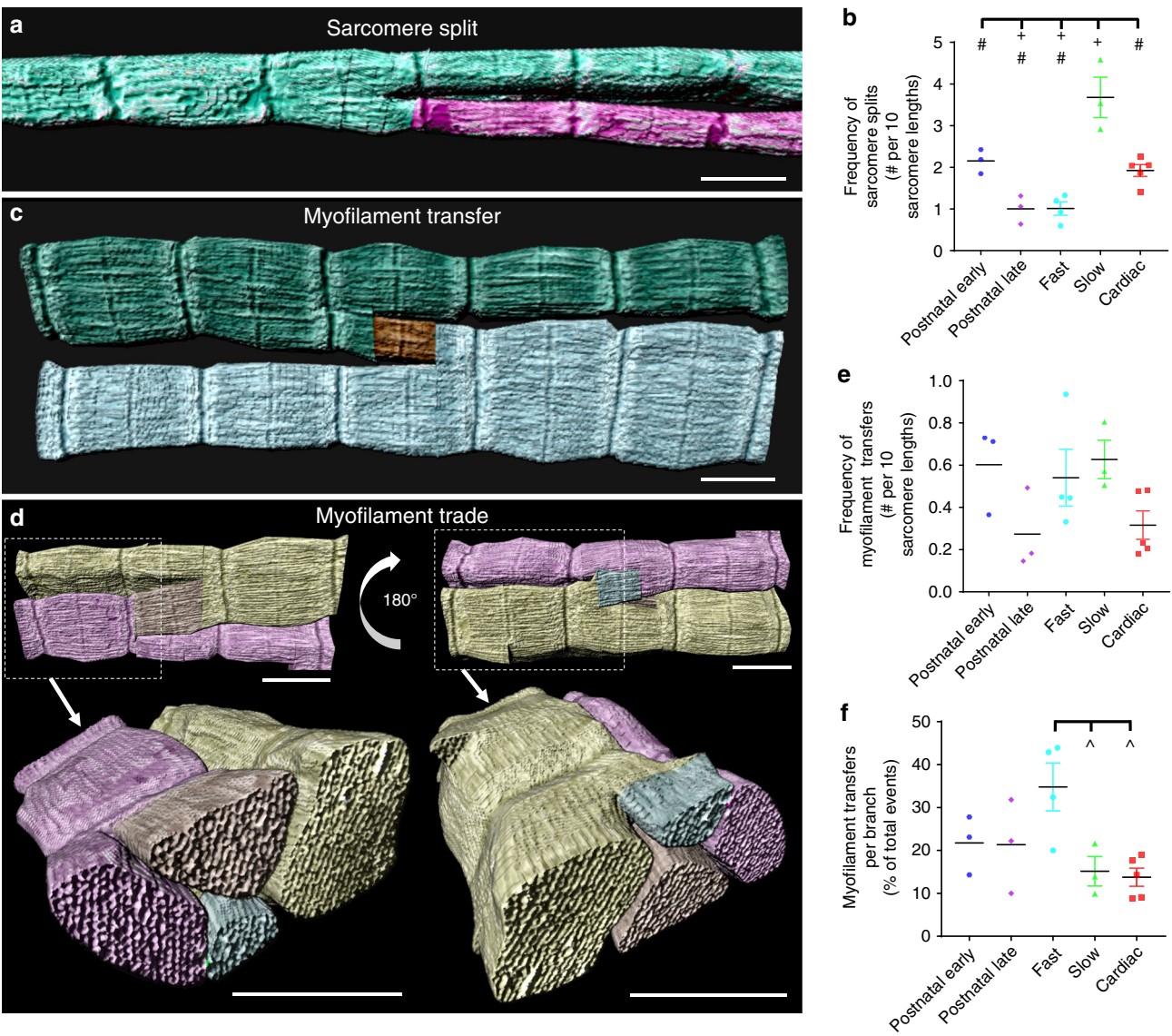

**Fig. 3 Types of sarcomere branching. a** 3D rendering of a sarcomere split. Individual colors represent different myofibrillar segments linked by branching sarcomeres. Representative of 1766 splits, 18 muscles, 16 mice. **b** Frequency of sarcomere splits. **c** 3D rendering of a transfer of myofilaments from one sarcomere to another. **d** 3D renderings of a transfer where myofilaments are traded between two sarcomeres. **c**, **d** are representative of 525 transfers, 18 muscles, 16 mice. **e** Frequency of myofilament transfers. **f** Percentage of myofilament transfers relative to total sarcomere branches. N values for **b**, **e**, **f**: postnatal early—141 myofibrils, 1796 sarcomeres, 3 muscles, 3 mice; postnatal late—150 myofibrils, 1401 sarcomeres, 3 muscles, 3 mice; fast twitch—193 myofibrils, 3477 sarcomeres, 4 muscles, 3 mice; slow twitch—110 myofibrils, 1396 sarcomeres, 3 muscles, 3 mice; cardiac—167 myofibrils, 2114 sarcomeres, 5 muscles, 5 mice. Plus (+): Significantly different (one-way ANOVA, $P < 0.05$) from postnatal early. Pound sign (#): Significantly different from slow. Up arrow (ˆ): Significantly different from fast. Scale bars, 1 μm. Values: mean ± SE.

focused more closely on the physical means by which branching occurred (Fig. 3). The most common type of sarcomere branching occurred through a splitting mechanism (Fig. 3a, b, Supplementary Movie 6), where some of the myofilaments within a single sarcomere separated from the rest resulting in two distinct myofibrillar structures. No specific structures appearing to cause separation of the contractile structures were observed directly at the site of sarcomere splitting, suggesting that splitting may be the result of more distal forces acting on the myofibril[30] or that cytoskeletal structures not visualized by our FIB-SEM approach may be involved. Differences in the frequency of sarcomere splitting among muscle types were nearly identical to the differences in sarcomere branching where slow-twitch fibers had the highest frequency and late postnatal and fast-twitch fibers had the lowest frequency (Fig. 3b), further suggesting fiber-type specific regulation of these events.

The second, less frequent type of sarcomere branching occurred through a transfer mechanism, where some of the myofilaments within a single sarcomere separated from the rest and joined an adjacent, parallel sarcomere (Fig. 3c). Thus, unlike sarcomere splitting, myofilament transfers did not result in additional myofibrillar structures. Instead, sarcomere transfers redistributed myofilaments between adjacent sarcomeres, thereby altering the size and shape of the resulting myofibrillar structures in addition to providing a direct force generating link between them. Myofilament transfers were also observed to occur in a bidirectional manner where myofilaments appeared to be traded between two adjacent sarcomeres (Fig. 3d, Supplementary Movie 7). There were no differences in the frequency of myofilament transfers between muscle types (Fig. 3e). However, in the fast-twitch muscle, a greater proportion of the total number

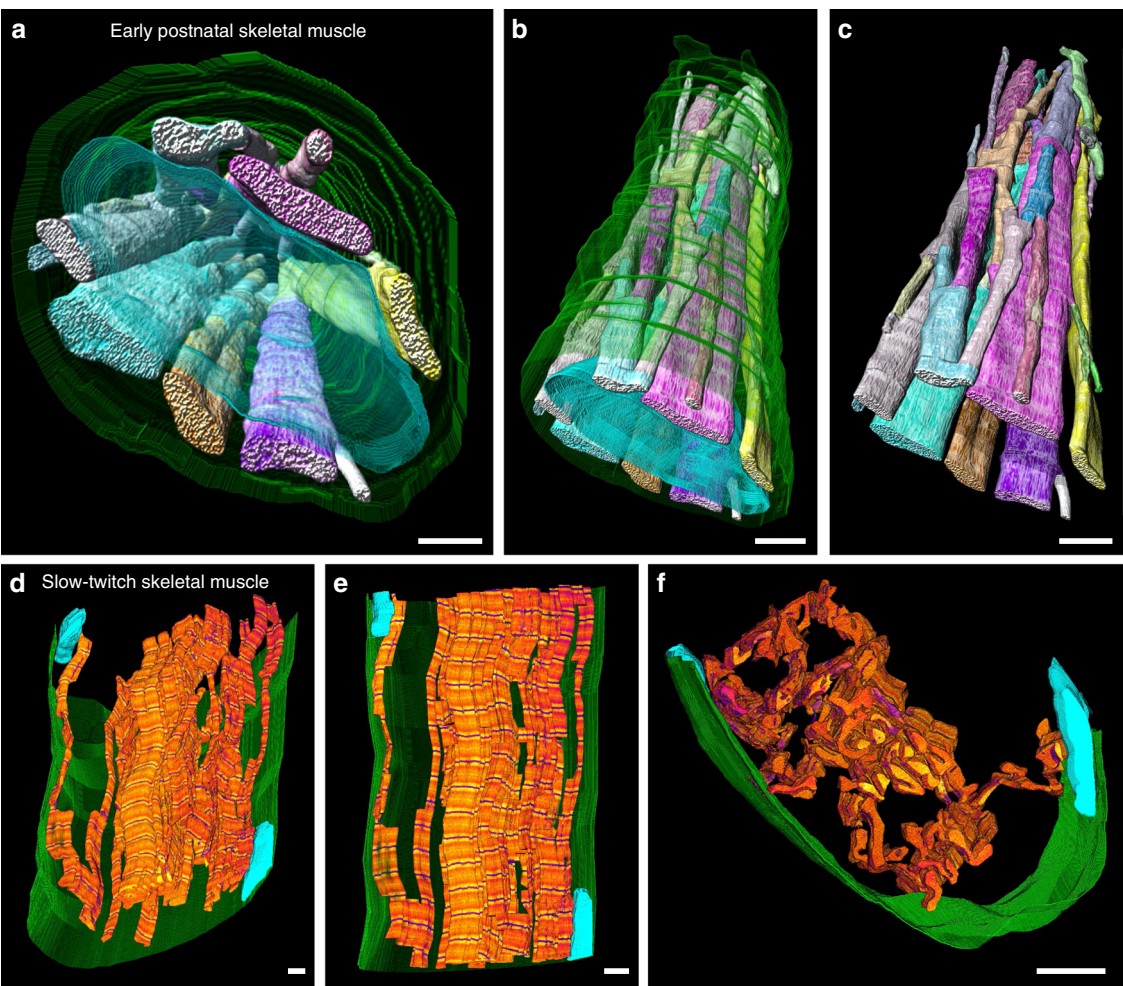

**Fig. 4 The myofibrillar matrix is directly connected across the entire width of the muscle fiber. a** 3D rendering of directly connected sarcomeres of the myofibrillar matrix throughout an early postnatal muscle fiber. Myofibrillar segments: various colors, nuclei: translucent cyan, cell boundary: translucent green. **b**, **c** 3D renderings of different perspectives highlight the connectivity of the myofibrillar matrix across the entire cell in the early postnatal stage. **d** 3D rendering of directly connected sarcomeres of the myofibrillar matrix across a slow-twitch muscle fiber. Contractile structures: orange, z-disks: dark blue, nuclei: cyan, cell boundary: green. For visualization purposes, not all connected sarcomeres are shown, only a minimal path of connectivity across the cell appears. **e**, **f** 3D renderings of different perspectives highlight the connectivity of the slow-twitch myofibrillar matrix across the entire cell. Images representative of three datasets from three mice. Scale bars, 2 μm. See also: Supplementary Movies 8 and 9.

of sarcomere branches (34.8 ± 5.6%, $n = 4$ muscle volumes, 193 myofibrils, 3477 sarcomeres) were due to myofilament transfers (Fig. 3f) compared to either slow-twitch (15.2 ± 3.4%, $n = 3$ muscle volumes, 110 myofibrils, 1396 sarcomeres) or cardiac muscles (13.8 ± 2.1%, $n = 5$ muscle volumes, 167 myofibrils, 2114 sarcomeres), suggesting that the direct force generating link between parallel myofibrillar structures created by myofilament transfers may offer a functional advantage and be upregulated relative to sarcomere splitting in the high force generating fast-twitch fibers.

**Sarcomere branching unites the entire cell**. It is well understood that myofibrillar structures are linked along the entire length of the muscle fiber providing a direct, longitudinal force generation and transmission pathway[6,7]. To determine the extent to which the myofibrillar matrix provides a pathway for force transmission along the lateral axis, we evaluated whether sarcomere branching could also directly link the contractile apparatus across the entire width of the muscle cell. In early postnatal muscle fibers, which still have centralized nuclei, we found that the frequent sarcomere branches resulted in a united myofibrillar network across both the

length and width of the muscle within our field of view (Fig. 4a–c, Supplementary Movie 8). These data suggest that a singular myofibrillar matrix directly connecting the entire contractile apparatus throughout the whole muscle cell is already formed at birth even before the nucleus is moved to cell periphery[31]. To examine whether the myofibrillar matrix is also directly linked across the entire width of a mature muscle fiber, we further assessed the connectivity of slow-twitch fibers, which, due to their high branching frequency (Fig. 2c), would be most likely to extend across the entire width of the muscle within the field of view of our datasets. Indeed, by tracing myofibrillar connectivity through z-disks as described for Figs. 1 and 2, but only following the minimal path of connectivity rather than following every sarcomere branch, we were able to demonstrate a directly connected myofibrillar pathway across the entire width of a slow-twitch muscle fiber (Fig. 4d–f, Supplementary Movie 9). Thus, the myofibrillar matrix is also directly connected across the entire length and width of the adult muscle cell. These results suggest that mammalian striated muscle cells form a single, unified myofibrillar matrix, which provides a direct pathway for both lateral and longitudinal active force generation throughout the cell.

## Discussion

The forces generated by actin–myosin crossbridge cycling within each skeletal muscle sarcomere must be transmitted to the muscle tendon in order to produce movement of the skeletal system, and it has long been accepted that force is transmitted longitudinally along the length of the muscle through the sarcomeres in series that form individual, parallel myofibrils[6,7]. Over the past four decades, there has been strong support for the importance of perpendicular cytoskeletal connections among parallel sarcomeres, the cell membrane, and the extracellular matrix in providing a means for lateral passive force transmission directly linked to the tendon[15,32], but the precise mechanisms of lateral force transmission during muscle contractions are not understood[15–18]. Here, we establish a unified myofibrillar matrix connected by branching sarcomeres across the entire length and width of the muscle cell, which provides a direct pathway for both longitudinal and lateral active force transmission from within individual sarcomeres to the tendon. Thus, it appears that both the myofibrillar matrix and cytoskeletal connections, such as desmin, work in tandem[10] to transmit both the active forces produced by actin–myosin crossbridge cycling and passive, viscoelastic forces laterally within the muscle cell.

Muscle fiber cross-sectional area increases several fold during postnatal muscle development[29], and this is achieved in large part by both an increase in the diameter of individual sarcomeres as well as an increase in the number of sarcomeres in parallel[33]. Based on previous 2D studies[33,34], splitting of single myofibrils was suggested to contribute to the muscle growth process by generating additional smaller myofibrils, which could then continue to grow. While we do not discount the role of sarcomere splitting in the muscle growth process, our analyses of the complete 3D structures of 20,000+ sarcomeres and 2400+ sarcomere branches across three developmental time points, three adult muscle types, and two species suggest that muscle growth and development are not the primary regulators of sarcomere branching. The rate of muscle growth in mice is greatest in the first month of life[29]. However, we found that 2-week-old mice have the lowest frequency of sarcomere branching of any muscle assessed (Fig. 2c), demonstrating that sarcomere branching is actually downregulated during periods of rapid muscle growth. Additionally, mature slow-twitch fibers are smaller than fast-twitch fibers and thus require less growth to reach maturity[29]. However, the frequency of sarcomere branching in the slow-twitch fibers is 2.8-fold higher than in the fast-twitch fibers (Fig. 2c), again demonstrating that the frequency of sarcomere branching does not correlate with muscle growth. Further, sarcomere branching frequency was similar between the late postnatal and mature, fast-twitch muscles (Fig. 2c), despite the large growth that occurred between these points. Conversely, sarcomere branching occurred 3.7-fold more frequently in mature, slow-twitch muscle than in the late postnatal muscle. Thus, we found that developmental regulation of sarcomere branching frequency is more closely related to the differentiation of muscle fiber type rather than as a means to increase sarcomere size or numbers.

Our demonstration of a singular myofibrillar matrix uniting the contractile apparatus throughout the entire muscle cell results in a very different understanding of muscle contractile function than is currently recognized. Undoubtedly, much of our knowledge of muscle contractile function has been guided by our understanding of muscle structure[4,5]. Indeed, the idea that a muscle cell is comprised of many individual myofibrils has resulted in thousands of studies on so-called isolated myofibrils over the past seven decades[35–41]. Although several key insights into muscle contractile function have been derived from experiments on isolated myofibrils[36,37,39], our demonstration of a

unified myofibrillar matrix here brings into question exactly what is being studied with these isolated myofibril preparations. Isolated myofibril studies are typically performed on small, unbranched myofibrillar segments consisting of 10–30 sarcomeres in series[35–41] (25–75 μm in length[42]). While these appear as long, thin contractile structures under a microscope; in fact, they represent <1% of the length of the entire muscle cell if derived from a mouse lower hindlimb as used here (~10 mm in length[43–45] or ~4000 sarcomeres[42]) and even less if from a larger animal such as the commonly used rat, rabbit, or frog. The field of view of our FIB-SEM datasets approximated the lengths used in isolated myofibril studies (25–65 μm or ~10–25 sarcomere lengths), and we did observe some myofibrils that did not branch within our field of view, albeit in a very small proportion of both fast-twitch (3.9 ± 1.4%) and slow-twitch (0.6 ± 0.6%) myofibrils (Fig. 2b). While this small number of non-branching myofibrillar segments is likely due to the relatively small field of view permitted by our imaging approach, these longest non-branching segments are also likely to be those used for isolated myofibril studies in which a single myofibrillar structure is typically chosen from among many in a dish. Additionally, the preponderance of isolated myofibril studies have been performed on fast-twitch muscles in which sarcomeres branch far less frequently than slow-twitch muscle, thereby making it easier to find longer non-branching segments. Indeed, it is difficult to find published images of isolated myofibrils from slow-twitch muscles, and those that are available show clear branching structures[40,41]. Moreover, the lack of functional studies on slow-twitch isolated myofibrils is puzzling considering that muscle fiber-type differences have been investigated thoroughly for most aspects of basic muscle biology. The highly branching nature of the slow-twitch myofibrillar matrix shown here suggests it would be difficult to find long, non-branching myofibrillar segments and provides an explanation for the dearth of slow-twitch isolated myofibril studies. Finally, we are unaware of any specific evidence of a single myofibril running the entire length of a mammalian muscle fiber. The highly connected nature of the myofibrillar matrix uniting the contractile apparatus across the entire length and width of the muscle cell both at birth and in mature muscles, as shown here (Fig. 4), suggests that there are no individual myofibrils within mammalian muscle. Instead, we propose that there are only sarcomeres in series and sarcomeres in parallel, which are all directly connected through sarcomere branches.

Branching of sarcomeres occurs in both directions along the longitudinal axis of muscle fibers, and branches in opposing directions often occur along the same myofibrillar segment, resulting in an interconnected zig-zagging myofibrillar structure (e.g., see annotated sarcomere branches in Fig. 1a). The high frequency and bidirectionality of sarcomere branching together with the changes in branching frequency observed throughout postnatal development suggest that sarcomere branching within the myofibrillar matrix may be dynamic in nature. Unfortunately, myofibrillar matrix dynamics cannot be assessed in the fixed muscle samples required for FIB-SEM imaging. Nonetheless, all sarcomeres in the muscle interior were observed to be directly connected in series with other sarcomeres whether branching occurred or not. In other words, no partially disconnected or free floating sarcomere or myofilament ends were found which would be expected if they were moving between structures, thus, arguing against dynamic reconfiguration of whole sarcomeres. However, it is possible that dynamic transfer of individual myofilaments between adjacent sarcomeres could still occur as actin is not visible in our images and tracking of individual myosin filaments is difficult through our image volumes even with 10 nm pixel sizes. The static nature of our images also cannot resolve the

directionality of myofilament transfers, that is, which sarcomere is receiving and which sarcomere is donating, nor whether some of the observed sarcomere splits are due instead to a merging of two sarcomeres. Notwithstanding, the distinct differences in sarcomere connectivity among muscle types reported here suggest that a remodeling of the myofibrillar matrix does indeed occur during postnatal development and fiber-type differentiation.

Shifting the paradigm of muscle physiology to include the myofibrillar matrix as the basic contractile structure of the muscle cell provides a different perspective on fundamental questions related to human movement, cardiac and skeletal muscle development, and related pathologies. In addition to establishing a direct mechanism of lateral force transmission among sarcomeres within a muscle cell, we found that the connectivity of sarcomeres within the myofibrillar matrix is regulated by developmental stage and fiber type, suggesting that the specific network structure of the myofibrillar matrix may be integral for muscle contractile function. The content and structure of muscle cells is finely tuned to achieve a specific function[20], and similar to other organelles (mitochondria, lipid droplets, etc.[20]), we found that the myofibrillar connectivity is regulated according to cell type, suggesting that these structures may be related to functional capacity. For example, the greater reliance on myofilament transfers between parallel sarcomeres in fast-twitch muscle suggests that this mechanism of sarcomere branching may be advantageous to muscle cells designed for rapid, high force contractions, whereas the highly branched myofibrillar structure of slow-twitch muscle cells results in relatively greater sarcomere surface areas, which may facilitate interactions between myofilaments and sites of ATP production and calcium cycling. Certainly, human pathologies such as muscular dystrophies, autoimmune and neurological diseases, and aging are all associated with alterations in muscle fiber-type composition, decrements in muscle force production, and myofibrillar misalignment[46–50], and thus, the evaluation of myofibrillar matrix structures in models of pathology may elucidate novel mechanisms of muscle dysfunction.

## Methods

**Mice**. All procedures were approved by the National Heart, Lung, and Blood Institute Animal Care and Use Committee and performed in accordance with the guidelines described in the Animal Care and Welfare Act (7 USC 2142 § 13). Six- to eight-week-old C57BL6/N mice were purchased from Taconic Biosciences (Rensselaer, NY) and fed ad libitum on a 12-h light, 12-h dark cycle at 20–26 °C. Breeding pairs were setup, and progeny were randomly selected for each experimental group. Early postnatal mice were from P1 and late postnatal mice were from P14. Adult mice were 2–4 months of age. Animals were given free access to food and water and pups were weaned at P21. Due to difficulty using anogenital distance to reliably determine gender in P1 pups, we did not group mice depending on sex, but randomly used both male and female mice.

**Muscle preparation**. Mouse hindlimb and cardiac muscles were fixed and prepared for imaging as described previously[19]. Mice were placed on a water circulating heated bed and anesthetized via continuous inhalation of 2% isoflurane through a nose cone. Hair and skin were removed from the hindlimbs and the legs immersed in fixative containing 2% glutaraldehyde in 100 mM phosphate buffer, pH 7.2, in vivo for 30 min. For heart fixation, the chest cavity was opened, and cardiac tissue was perfusion fixed through the apex of the left ventricle by slowly pushing 2 ml of relaxing buffer (80 mM potassium acetate, 10 mM potassium phosphate, 5 mM EGTA, pH 7.2), followed by 2 ml of fixative solution through a syringe attached to a 30 G needle. After initial fixation, the gastrocnemius, soleus, and/or left ventricles were then removed, cut into 1 mm³ cubes, and placed into 2.5% glutaraldehyde, 1% paraformaldehyde, 120 mM sodium cacodylate, pH 7.2–7.4 for 1 h. After five, 3-min washes with 100 mM cacodylate buffer at room temperature, samples were placed in 3% potassium ferrocyanide, 200 mM cacodylate, 4% aqueous osmium on ice for 1 h, washed five times in bi-distilled $H_2O$ for 3 min, and incubated for 20 min in fresh thiocarbohydrazide solution at room temperature. Samples were then incubated for 30 min on ice in 2% osmium solution and washed five times in bi-distilled $H_2O$ for 3 min. Samples were next incubated overnight at 4 °C in 1% uranyl acetate solution, washed five times in bi-distilled $H_2O$ for 3 min, incubated in 20 mM lead nitrate, 30 mM aspartic acid, pH 5.5 at 60 °C for 20 min, and washed five times in bi-distilled $H_2O$ at room

temperature for 3 min. Samples were then incubated sequentially in 20%, 50%, 70%, 90%, 95%, 100%, and 100% ethanol for 5 min each, incubated in 50% Epon solution, 50% ethanol for 4 h, and incubated in 75% Epon, 25% ethanol overnight at room temperature. Epon solution was prepared as a mixture of four components: 11.1 ml Embed812 resin, 6.19 ml DDSA, 6.25 ml NMA, and 0.325 ml DMP-30. Epon solution was thoroughly mixed with a magnetic stir bar in a vacuum container. The following day, samples were incubated in fresh 100% Epon for 1, 1, and 4 h, sequentially. After removing excess resin using filter paper, the samples were placed on aluminum ZEISS SEM Mounts in a 60 °C oven for two days. Stubs were then mounted in a Leica UCT Ultramicrotome (Leica Microsystems Inc., USA) and trimmed with a Trimtool 45 diamond knife with a feed of 100 nm at a rate of 80 mm/s.

**Focused ion beam-scanning electron microscopy**. FIB-SEM images were acquired using a ZEISS Crossbeam 540 with ZEISS Atlas 5 software (Carl Zeiss Microscopy GmbH, Jena, Germany) and collected using an in-column energy-selective backscatter with filtering grid to reject unwanted secondary electrons and backscatter electrons up to a voltage of 1.5 kV at the working distance of 5.01 mm. FIB milling was performed at 30 kV, 2–2.5 nA beam current, and 10 nm thickness. Image stacks within a volume were aligned using Atlas 5 software (Fibics Incorporated) and exported as TIFF files for analysis.

**Image segmentation**. Raw FIB-SEM image volumes were rotated in 3D so that the XY images within the volume were of the muscle cell cross-section (e.g., Supplementary Fig. 1a). Myofibrillar structures were segmented using the interpolation feature within theTrakEM2 plugin in ImageJ. The cross-sectional shape of a single sarcomere in the first image of the volume (see labels in Supplementary Fig. 1a) was traced manually and then additional sarcomere cross-section tracing was performed sequentially for each connected sarcomere in series at the H-zones and Z-disks, which were readily apparent based on contrast changes within the grayscale images (e.g., see annotations in Fig. 1a). Interpolation of H-zone and Z-disk cross-sections resulted in accurate representations of the full sarcomere structures (Supplementary Movie 1). When sarcomeres branched during tracing, cross-sections were traced immediately before and after the branching point to ensure accurate interpolation of sarcomeric structures. At sarcomere branch points, the previous segment was continued along one part of the branch, while a new segment was created for the other part allowing for multi-color representation of each myofibrillar segment within the connected myofibrillar matrix (e.g., Fig. 1). Accuracy of complete myofibrillar segmentations was assessed by overlaying the segmented structures on the raw image files to ensure no errors in connectivity or sarcomere structure were made.

**Image analysis**. Raw FIB-SEM image volumes were rotated in 3D so that the XY images within the volume were of the muscle cell cross-section (e.g., Supplementary Fig. 1a). Every sarcomere in parallel in the first image within the dataset was then numbered sequentially for tracking purposes (e.g., Supplementary Fig. 1a). Only sarcomeres whose serial structures stayed within the field of view over the entire dataset were used to calculate the frequency of sarcomere branching, splitting, or myofilament transfers. Beginning with the first, numbered image in the dataset, each sarcomere and its subsequent sarcomeres connected in series through the z-disks were tracked along the length of the muscle and the number of times the sarcomeres split or merged and the number of times myofilaments were transferred to or from adjacent sarcomeres in parallel were counted. When sarcomere splits or myofilament transfers occurred, tracking proceeded down the length of the muscle with the portion of the myofibrillar structure, which best maintained the same relative location within the muscle fiber (i.e., distance from the cell membrane) as the original numbered sarcomere from the first image.

**Image rendering**. Movies of 3D renderings of myofibrillar structures were generated in Imaris 9.5 using the horizontal and vertical rotation animations and/or the clipping tool. Pictures of 3D renderings were created either using Imaris or the Volume Viewer plugin in ImageJ.

**Statistical analysis**. Quantitative data was assessed in Excel 2016 (Microsoft) and Prism 7 (GraphPad) was used for all statistical analyses. All comparisons of means between early postnatal, late postnatal, fast-twitch, slow-twitch, and cardiac structures were performed using a one-way analysis of variance with a Tukey's HSD (honestly significant difference) post hoc test. A $p$ value $< 0.05$ was used to determine statistical significance.

**Reporting summary**. Further information on research design is available in the Nature Research Reporting Summary linked to this article.

## Data availability
The datasets generated during and/or analyzed during the current study are available from the corresponding author on reasonable request. Source data are provided with this paper.

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

## Acknowledgements

We thank Brian Caffrey (University of British Columbia), Sriram Subramaniam (University of British Columbia), and Luigi Ferrucci (National Institute of Aging) for providing access to the raw human muscle FIB-SEM datasets from the Baltimore Longitudinal Study of Aging. This work was supported by the Division of Intramural Research of the National Heart Lung and Blood Institute and the Intramural Research Program of the National Institute of Arthritis and Musculoskeletal and Skin Diseases.

## Author contributions

B.G. and Y.K. prepped tissues for imaging. B.G., Y.K., E.L., and C.K.E.B. designed and E.L. and C.K.E.B. performed imaging experiments. T.B.W. and B.G. designed and performed image analysis and created figures. T.B.W. and B.G. wrote the manuscript.

## Competing interests

The authors declare no competing interests.
