## [Peer Review File · Nature Communications]

Reviewers' Comments:

Reviewer #1:

Remarks to the Author:

I find this paper interesting, well conducted and well described. However, its main finding (branching of myofibrils occurs much more frequently than assumed by most so far) is somewhat limited and I am hoping the authors can take their work further.

I am concerned that the branching could merely reflect catching the muscle fiber growing in cross-sectional area, a process that is well-known to occur through splitting of myofibrils (and subsequent growth of the mfs.). It would be important to confirm that splitting of myofibrils occurs at the same frequency as reported here in more mature mice (exceeding the maximal age range used here, 2-4 months) and also if possible in muscles that undergo atrophy (e.g., through hind limb suspension, or denervation). Is there a correlation between where a branch-point occurs and the width of the myofibril just before the branch point?

This study could be made more interesting by knowing more about how the branching relates to changes in 3D arrangement of the T-tubular and SR systems and the network of mitochondria.

If branching of myofibrils occurs as frequently as shown here (2-4 branches per 10 sarcomeres) how can those who perform mechanical studies on single myofibrils obtain long unbranched myofibrils for their studies?

How does branching relate to cytoskeletal connections (desmin)? If separation between adjacent myofibrils is enlarged due to local branching do connections increase in length or might connections then be absent?

The functional significance of branching and myofilament transfer needs more in-depth discussion. Why would myofilament transfer lead to greater structural support for high force contractions (line 343)? Could one make the argument that since branching might lead to variation in the total cross-sectional area of myofibrils, branching leads to instability? And how does the functional significance of branching/myofilament transfer relate to the mechanical role of desmin connections?

Considering the high uncertainty around the functional significance of branching/myofilament transfer, the speculation that it presents a therapeutic target (end of discussion) is somewhat of a stretch. Please remove or be more convincing in your arguments.

Reviewer #2:

Remarks to the Author:

This interesting paper shows in beautiful 3-dimensional detail how myofibrils in striated muscles can split longitudinally and can sometimes merge with an adjacent myofibril. However, much of this is not entirely new. Under normal circumstances, the number of muscle fibres in a particular muscle is thought to stay constant after early adulthood, and further development of muscles to produce larger tensions has been known for many years to be due to lateral myofibril growth followed by splitting of the myofibrils into two when they reach a certain size. The two 'daughter' myofibrils can then continue to grow. That the splitting might occasionally contain a discontinuity so that longitudinal tracks of sarcomeres can start in one myofibril and then occasionally transfer to the other of the two 'daughters' does not seem surprising. As far as I am aware the local molecular factor that drives the splitting is not known.

In summary, the conclusions of the authors do not seem to me to be surprising. However, they

have imaged the splitting and filament transfer between myofibrils with great clarity using the relatively new FIB-SEM technique, they have produced some beautiful images, and for the first time

they have been able to quantify the relative abundance of splits and transfers in a good range of muscle types and at various stages of development.

So, I like this paper and it adds substantially to our understanding of what is going on during fibre development. My only query is whether this is the appropriate journal for it, rather than an anatomy or muscle Journal.

Specific comments:

Line 66: understanding of how cellular.....

Line 258:Over the last four decades.....

Line 260:and the extracellular matrix....

Line 293: The word intensity to me has a specific meaning and does not apply to contractions. Perhaps use 'lower strength contractions' or 'lower contractile forces' instead?

Reviewers' comments:

Reviewer #1 (Remarks to the Author):

I find this paper interesting, well conducted and well described. However, its main finding (branching of myofibrils occurs much more frequently than assumed by most so far) is somewhat limited and I am hoping the authors can take their work further.

We thank the reviewer for their interest in our work as well as for their constructive suggestions. However, the primary finding here is not that branching occurs more frequently than previously thought, but rather, it is the sum of those branches that results in a completely connected, singular contractile apparatus. In other words, there are no individual myofibrils running the entire length of the cell within mammalian muscle as is the common assumption. The questions and comments by reviewer below have helped point out where we lacked adequate clarity of these points in the original submission which we have now addressed in the revised manuscript and below.

I am concerned that the branching could merely reflect catching the muscle fiber growing in cross-sectional area, a process that is well-known to occur through splitting of myofibrils (and subsequent growth of the mfs.). It would be important to confirm that splitting of myofibrils occurs at the same frequency as reported here in more mature mice (exceeding the maximal age range used here, 2-4 months) and also if possible in muscles that undergo atrophy (e.g., through hind limb suspension, or denervation). Is there a correlation between where a branch-point occurs and the width of the myofibril just before the branch point?

We thank the reviewer for the suggestion to more clearly demonstrate that sarcomere branching is not simply part of the muscle fiber growth process. While we do not discount a role for sarcomere branching in muscle development as previously described, we believe our 3D analysis of sarcomere branches across five muscle types at three developmental time points suggests a much larger role for sarcomere branching which is ubiquitous across all vertebrate muscles regardless of developmental or growth stage. Our reasoning for this is clarified below and further, more specific discussion of this point has now been added to the main text.

- 1) The rate of muscle growth in mice is greatest in the first month of life (Rowe and Goldspink, J Anat, 1969). However, we show that 2 week old mice have the lowest frequency of sarcomere branching of any timepoint assessed. This is inconsistent with muscle growth as the primary regulator of sarcomere branching.**
- 2) Mature slow-twitch fibers are smaller than fast-twitch fibers and thus require less growth to reach maturity (Rowe and Goldspink, J Anat, 1969). However, the frequency of sarcomere branching in slow-twitch fibers is 2.8-fold higher than in fast-twitch fibers. This is inconsistent with muscle growth as the primary regulator of sarcomere branching.**
- 3) Many sarcomere branches occur through myofilament transfers between adjacent sarcomeres. Myofilament transfers do not provide a means for muscle growth as one sarcomere gets smaller while another gets bigger.**

- 4) Muscle fiber diameter increases until ~9 weeks of age in mice and then stabilizes (Rowe and Goldspink, *J Anat*, 1969). Thus, the four month old mice assessed here likely have the mature muscles desired for analysis.
- 5) To further demonstrate that a unitary myofibrillar matrix is the normal structure of stable, mature muscle cells, we have now assessed the myofibrillar structures from adult muscles from humans (age <40) which were part of the Baltimore Longitudinal Study of Aging (Caffrey et al. *J Struct Biol*, 2019). Though the human datasets we obtained were smaller in volume than those we collected for mice, every myofibril we assessed had at least one sarcomere branch, and the sarcomere branching frequency in these human fast-twitch muscles was as high as, if not higher than in our fast-twitch mouse muscles. These data are now included as Supplemental Figure 2 and further suggest that frequent sarcomere branching is the normal structure of all mammalian skeletal muscle cells, not just those which are growing.
- 6) There does not appear to be a correlation between myofibril width and where a branch occurs as sarcomere branching frequency is higher at birth than at either two weeks of age or in adult fast-twitch fibers which have larger myofibril widths than at birth.

This study could be made more interesting by knowing more about how the branching relates to changes in 3D arrangement of the T-tubular and SR systems and the network of mitochondria.

We thank the reviewer for this suggestion, we have now provided 3D renderings of the relationship between the myofibrillar matrix, mitochondria, and the sarcotubular system showing that sarcomere branching does not appear to alter the normal relationships between these organelles and the contractile apparatus. These data are included as Supplemental Videos 3 and 4.

If branching of myofibrils occurs as frequently as shown here (2-4 branches per 10 sarcomeres) how can those who perform mechanical studies on single myofibrils obtain long unbranched myofibrils for their studies?

We thank the reviewer for this question and agree that further discussion of this issue is warranted in the manuscript and has now been added. Isolated myofibril studies are typically performed on myofibrillar segments of 10-30 sarcomeres in series and very rarely on segments of 50+ sarcomeres in series after harsh mechanical (i.e. homogenization) and/or chemical (e.g. detergent) treatments. While these isolated myofibrils appear long, they are very short compared to the length of the muscle fiber from which they belong to. Mouse hindlimb muscles as used in our study are ~10 mm in length. Assuming a sarcomere length of 2.5 μm and no branching, a single myofibril would be ~4000 sarcomeres in length or more than a hundred times longer than the typical isolated myofibril. Additionally, many of the classical isolated myofibril studies were performed on frog or rabbit muscles which are even longer than in the mouse and would make the length of the isolated myofibrils relatively even smaller. Thus, the "isolated myofibrils" reported on in the literature are hardly representative of individual myofibrils running the entire length of the cell as our historical understanding of myofibril structure would suggest. We are unaware of any specific evidence of a single myofibril running the entire length of a mammalian muscle fiber.

Additionally, functional isolated myofibril studies have been performed largely on fast-twitch muscle which branches much less frequently than slow-twitch muscle, thereby making it easier to find longer non-branching segments. Indeed, we find that a small number of fast-twitch myofibrils (<5%) do not branch within the 10-25 sarcomere lengths observed in our image volumes making these segments likely to be those chosen for “isolated myofibril” studies after tissue homogenization. However, it is difficult to find images of isolated slow-twitch muscle myofibrils in the literature. For the two examples we were able to find from adult human triceps (Heizmann et al. Eur J Biochem, 1983) and chicken anterior latissimus dorsi (Shafiq et al. Muscle and Nerve, 1984) muscles, branching is apparent in all images (images included below with red arrows highlighting branches). It is important to note that even these two studies were not functional in nature, but rather, they focused on the composition of slow-twitch myofibrils. Most aspects of basic muscle biology have been compared between the different muscle fiber types begging the question as to why isolated myofibril function has been largely assessed in only fast-twitch muscle. Our data showing the highly branched nature of the slow-twitch muscle contractile apparatus suggests it would be very difficult to find the long, unbranched segments needed for functional isolated myofibril studies in these muscles. Thus, the lack of study on slow-twitch isolated myofibrils may be explained by our data.

Fig. 6. Localization of type I protein (a – d) by the indirect immunofluorescence technique in isolated human myofibrils; myofibrils incubated with preimmune serum (e,f). The photographs on the left show the phase-contrast pictures (a, c, e) and on the right the corresponding fluorescence (b, d, f) of the same myofibril (a, b). Magnification $\times 3000$ and (c – f) $\times 1500$. Note that not all myofibrils are stained (c, d, on the right)

How does branching relate to cytoskeletal connections (desmin)? If separation between adjacent myofibrils is enlarged due to local branching do connections increase in length or might connections then be absent?

We thank the reviewer for this insightful question. We believe sarcomere branching and cytoskeletal connections such as desmin likely work in tandem to transmit force within a muscle cell. Desmin and other cytoskeletal proteins have long been known to play a role in force transmission by linking together adjacent sarcomeres in parallel as well as sarcomeres to the cell membrane. Our data show that sarcomeres can also be linked across the entire width of the cell through sarcomere branches. Thus, both sarcomere branching and cytoskeletal connections provide a force transmission link across the width of the muscle cell. However, cytoskeletal connections are passive force transmitters as they do not generate force themselves, whereas sarcomere branching provides a pathway for direct transmission through the force generating machinery. Additionally, cytoskeletal connections provide a link from sarcomeres to the cell membrane which the myofibrillar matrix does not. We have now added these points to our discussion.

Separation between adjacent myofibrils does not appear to be enlarged due to sarcomere branching as all contractile structures observed remain tightly packed together (Supplemental Video 1). The larger gaps between myofibrillar segments in Figure 1 are actually filled with other myofibrillar structures which were not directly connected within our field of view. If anything, sarcomere branching may result in shorter cytoskeletal connections in the instances when the branch point occurs directly before the z-disk.

The functional significance of branching and myofilament transfer needs more in-depth discussion. Why would myofilament transfer lead to greater structural support for high force contractions (line 343)? Could one make the argument that since branching might lead to variation in the total cross-sectional

area of myofibrils, branching leads to instability? And how does the functional significance of branching/myofilament transfer relate to the mechanical role of desmin connections?

We agree that the functional significance of the myofibrillar matrix is of great interest. We have chosen to focus our additional discussion on the role of the myofibrillar matrix in providing a direct means for lateral transmission of active forces within the muscle cell as that is the function most directly explained by our data. As mentioned in the response to the question above, desmin is suggested to also provide a means for lateral force transmission but cytoskeletal connections can only do so passively whereas the myofibrillar matrix provides the mechanism for lateral transmission of active forces.

Considering the high uncertainty around the functional significance of branching/myofilament transfer, the speculation that it presents a therapeutic target (end of discussion) is somewhat of a stretch. Please remove or be more convincing in your arguments.

We have now removed the suggestion that sarcomere branching presents a therapeutic target and instead suggest that investigation into branching may provide new insights into the muscle dysfunction associated with these diseases.

Reviewer #2 (Remarks to the Author):

This interesting paper shows in beautiful 3-dimensional detail how myofibrils in striated muscles can split longitudinally and can sometimes merge with an adjacent myofibril. However, much of this is not entirely new. Under normal circumstances, the number of muscle fibres in a particular muscle is thought to stay constant after early adulthood, and further development of muscles to produce larger tensions has been known for many years to be due to lateral myofibril growth followed by splitting of the myofibrils into two when they reach a certain size. The two 'daughter' myofibrils can then continue to grow. That the splitting might occasionally contain a discontinuity so that longitudinal tracks of sarcomeres can start in one myofibril and then occasionally transfer to the other of the two 'daughters' does not seem surprising. As far as I am aware the local molecular factor that drives the splitting is not known.

We thank the reviewer for their interest in our work. We agree that demonstrations of the individual splits and merges themselves have been shown previously and we had included reference to several of these works. What was missing from the previous 2D investigations into sarcomere branching was an understanding of how these individual branches connect together in total to result in a singular, unified contractile apparatus rather than many individual pieces. This continued lack of understanding of the overall connectivity of muscle myofibrillar structures is reflected in the persistent study of so-called isolated myofibrils (see comment from Reviewer #1 above), the absence of accounting for splitting in any muscle cell force generation/transmission computational models, and the common statements in the literature that muscle cells are comprised of individual myofibrils which run the entire length of the cell (e.g. Roman et al. Nat Cell Biol, 2017). What we have aimed to better clarify

with this revision is that sarcomere branching links the entire contractile apparatus together across the length and width of the cell regardless of whether it is during early development (Figure 4a-c, Supplemental Video 8) or in mature skeletal muscle (Figure 4d-f, Supplemental Video 9). These data argue that there is no such thing as a single myofibril in mammalian skeletal muscles. Instead, there are sarcomeres in series and in parallel which make up the connected, non-linear, branching segments of a unified myofibrillar matrix.

As also outlined for Reviewer #1 above, we believe our data showing the full 3D connectivity of the muscle contractile apparatus argues against growth and response to high tension as being the primary drivers of sarcomere branching as suggested by previous 2D studies. We show that sarcomere branching frequency goes down and is lowest during the largest period of muscle growth (late postnatal development) and that the larger, higher tension generating fast-twitch fibers branch less frequently than slow-twitch fibers (Figure 2c). Thus, the muscle cell with the highest sarcomere branching frequency is the mature, slow-twitch fiber which is neither growing nor under high tension compared to the other muscle types.

In summary, the conclusions of the authors do not seem to me to be surprising. However, they have imaged the splitting and filament transfer between myofibrils with great clarity using the relatively new FIB-SEM technique, they have produced some beautiful images, and for the first time they have been able to quantify the relative abundance of splits and transfers in a good range of muscle types and at various stages of development.

We thank the reviewer for their appreciation of our images and large scale quantification of the 3D branching structures of the contractile apparatus within the muscle cell. The advantage provided by our 3D approach is that it allowed us to see how all the splits and transfers connect together in sum to yield a new view of how a muscle cell is built. We demonstrate the presence of a singular, mesh-like myofibrillar network which argues against the presence of any single myofibrils within the muscle cell. We have now further clarified these points by updating the figures, videos, and the text as outlined in the responses above.

So, I like this paper and it adds substantially to our understanding of what is going on during fibre development. My only query is whether this is the appropriate journal for it, rather than an anatomy or muscle Journal.

We thank the reviewer for their comments which have helped us to better clarify the primary finding of this work that the contractile apparatus of mammalian striated muscle cells is not made up of many individual myofibrils, but rather there is a singular myofibrillar matrix directly connected across the entire length and width of the muscle cell by branching sarcomeres.

Specific comments:

Line 66: understanding of how cellular..... **Thanks for catching these. Change made**

Line 258:Over the last four decades..... **Change made**

Line 260:and the extracellular matrix.... **Change made**

Line 293: The word intensity to me has a specific meaning and does not apply to contractions.

Perhaps use 'lower strength contractions' or 'lower contractile forces' instead?

This sentence has been removed in the revised manuscript

Reviewers' Comments:

Reviewer #1:

Remarks to the Author:

I remain enthusiastic about this work but have a few remaining comments.

1. The analysis of the human fibers adds to this work. However, I would like to see images in addition to the data graphs in supplementary figure 2.
2. line 326 states "... it is well known that these isolated myofibril preparations are not able to account for the forces generated by the intact muscle cell." This is a strong statement for which you do not cite any study or give any values. Additionally, the inference seems to be that to measure high specific-force levels, multiple myofibrils with branches would have to be studied. But how would branching result in higher force levels? A 3-D matrix of branching myofibrils could be a means to by-pass an area that is damaged and that generates low force levels. Alternatively it could be a way to ensure even shortening distances in different areas of the cell (and avoid thereby damage from occurring), but it is hard to see how it could give rise to specific forces that are higher than those achievable with isolated single myofibrils. Please expand.
3. The lack of functional studies on slow-twitch myofibrils (line 325) might have reasons that have nothing to do with the frequency of branching. One reason could be that the most commonly used animal model systems are not rich in slow-twitch muscle (in the mouse it is only the soleus muscle that is slow-twitch and even here only ~ 50% of the fibers are type I).
4. On page 390 you state that because the connectivity varies among muscle types and developmental stage it must be integral for muscle function. Why would that be? Can't it be an oddity that only has a minor function?
5. A more in-depth Discussion of the differences between active and passive force transmission (myofibrillar branching and desmin, respectively) would be helpful. Are they functionally similar or do they perform different roles?
6. In the revised manuscript please mark in the text where you made changes.

Reviewer #2:

Remarks to the Author:

In my view the authors have satisfactorily answered my queries and comments and I am happy for the paper to proceed.

REVIEWERS' COMMENTS:

Reviewer #1 (Remarks to the Author):

I remain enthusiastic about this work but have a few remaining comments.

We thank the reviewer for their continued interest in our work and additional constructive comments.

1. The analysis of the human fibers adds to this work. However, I would like to see images in addition to the data graphs in supplementary figure 2.

Thanks for this suggestion. We have now added an image panel to this figure.

2. line 326 states "... it is well known that these isolated myofibril preparations are not able to account for the forces generated by the intact muscle cell." This is a strong statement for which you do not cite any study or give any values. Additionally, the inference seems to be that to measure high specific-force levels, multiple myofibrils with branches would have to be studied. But how would branching result in higher force levels? A 3-D matrix of branching myofibrils could be a means to by-pass an area that is damaged and that generates low force levels. Alternatively it could be a way to ensure even shortening distances in different areas of the cell (and avoid thereby damage from occurring), but it is hard to see how it could give rise to specific forces that are higher than those achievable with isolated single myofibrils. Please expand.

Due to the lack of references, we have now removed the statement in question.

3. The lack of functional studies on slow-twitch myofibrils (line 325) might have reasons that have nothing to do with the frequency of branching. One reason could be that the most commonly used animal model systems are not rich in slow-twitch muscle (in the mouse it is only the soleus muscle that is slow-twitch and even here only ~ 50% of the fibers are type I).

We agree that there are generally fewer slow-twitch fibers in many animal models. However, there have still been hundreds of studies on other aspects of muscle biology done on slow-twitch muscle from mice, rats, rabbits (soleus is 98% type I), and even frogs. The procedure for isolating myofibrils would not seem to make it more difficult to assess slow-twitch fibers than other types of muscle biology experiments. In fact, the general grind it up and choose the best looking one approach would seem to make it easy to pick even low abundance slow-twitch myofibril segments which are well-known to be structurally distinct from fast-twitch myofibrils beyond their branching frequency (z-disk width, sarcomere cross-sectional shape, etc.). Thus, we still believe it is puzzling that there are not any studies done on slow twitch myofibrils and that the high frequency of sarcomere branches may offer an explanation.

4. On page 390 you state that because the connectivity varies among muscle types and developmental stage it must be integral for muscle function. Why would that be? Can't it be an oddity that only has a minor function?

While we find that myofibrillar connectivity is a fundamental component of myofibril structure across all cell types and development stages, the precise function of myofibrillar connectivity remains

unclear. Due to the widely described fiber type specific fine-tuning of other muscle cell structures for functional purposes, it is logical to hypothesize that myofibrillar branching is similar, though further testing will be required in order to make a definitive statement. Therefore, we have modified the text, and the statement from “is integral” to “may be integral” to muscle function.

5. A more in-depth Discussion of the differences between active and passive force transmission (myofibrillar branching and desmin, respectively) would be helpful. Are they functionally similar or do they perform different roles?

We have now included additional details to clarify the differences between active (actin-myosin crossbridge cycling) and passive (viscoelastic) forces in the opening paragraph of the discussion.

6. In the revised manuscript please mark in the text where you made changes.

Thank you for this suggestion. We have submitted a tracked change version of the manuscript.

Reviewer #2 (Remarks to the Author):

In my view the authors have satisfactorily answered my queries and comments and I am happy for the paper to proceed.

We thank the reviewer for their feedback which has helped to improve our manuscript.